# The Hierarchy of Controls as an Approach to Visualize the Impact of Occupational Safety and Health Coordination

**DOI:** 10.3390/ijerph19052731

**Published:** 2022-02-26

**Authors:** Jeppe Z. N. Ajslev, Jeppe L. Møller, Malene F. Andersen, Payam Pirzadeh, Helen Lingard

**Affiliations:** 1The National Research Centre for the Working Environment, 2100 Copenhagen, Denmark; jlm@nfa.dk; 2Independent Researcher, 2100 Copenhagen, Denmark; kontakt@malenefriisandersen.dk; 3Department of Property, Construction and Project Management, RMIT University, Melbourne, VIC 3000, Australia; payam.pirzadeh@rmit.edu.au (P.P.); helen.lingard@rmit.edu.au (H.L.)

**Keywords:** OSH coordinator, OSH professional, construction industry, observation study, occupational safety and health

## Abstract

Occupational safety and health (OSH) in construction work continues to be a problematic issue, and OSH coordinators are a pivotal initiative for improving this in the EU. However, no studies on the impact of (OSH) coordinators in construction exists. This study conceptualizes the *hierarchy of controls* (HOC) as a means for visualizing and evaluating the impact of OSH coordinators’ work. The study engages with a large observational material based on fieldwork notes from 107 days of observations with 12 successful OSH coordinators. The analysis shows that during the 107 observation days, the coordinators implemented 280 OSH measures and were prevented 71 times from implementing measures. Most of the implemented measures were in the administrative (53.6%) and engineering (35%) controls. This may provide part of the explanation of why an increasing focus on OSH coordination has not translated into improved OSH outcomes in construction. The study contributes with insights for OSH coordinators and professionals seeking to improve the visibility and legitimacy of their work. In addition, it may be beneficial to organizations interested in ensuring the effectiveness of their organizational OSH practices. The study also creates foundations for more research-based practices, education, and professionalization of OSH coordinators as a profession.

## 1. Introduction

Occupational illness and accidents in the European Union cost more than EUR 475 billion each year [1]. Since the 19th century [2], the construction industry has been, and continues to be, one of the most hazardous industries to work in. Occupational safety and health (OSH) threats, such as accidents and physically exerting work, still challenge management and OSH initiatives today [3,4,5]. These observations are emphasized by Eurostat, showing that 716 fatalities and 371,732 nonfatal accidents were registered in construction in 2016, and that the fatality rate in construction was 3.4 times higher than the EU average for 2016, and 1.9 times higher for nonfatal accidents [6].

Many OSH risks can be attributed to a number of well-known traits of construction work, such as the temporal character of the workplaces, the often multiple employers and professional groups working at the same time and in the same space, time pressure, competition, manual handling of heavy objects, manipulation of large objects, and work on different vertical levels [7,8,9].

One particularly important political initiative in seeking to improve OSH in the construction industry is health and safety coordination (OSH coordination), which is undertaken by the OSH coordinators—a particular type of OSH professional [10,11] who represents the client in matters of OSH throughout the project. Since its introduction in EU legislation through directive 92/57/EEC in 1992, OSH coordination has been pivotal for political initiatives to improve health and safety among construction workers. In brief, the OSH coordinator is appointed to coordinate OSH at sites with more than one employer present on behalf of the client [12]. Among other things, the OSH coordinator must ensure that all risks in the construction project are minimized during design, that employers apply the general prevention principles in their workspaces, that employers cooperate appropriately in matters of OSH, to conduct safety meeting and inspections, and that the shared workspace, pathways, and facilities outside the individual contractors’ workspaces are safe. Even though this politically emphasized area has existed for almost 20 years, little research has investigated the actual impact and quality of OSH coordination.

### Existing Research on OSH Coordination

One such rare study was conducted by Rubio et al. [13], who showed that OSH coordinators report that construction employers, as well as workers, often do not comply to safety directions [13,14], which poses an inherent barrier to improved safety. Research shows that OSH professionals, in general, struggle to gain traction, status, and organizational support in their efforts to improve OSH [15]. Recent research on OSH coordinators also points out that strategies of transferring knowledge into practice in organizations is perhaps the most fundamental competences of OSH coordinators [16].

While these recent studies indicate an increasing research interest on OSH coordination, no studies investigate to which extent OSH coordinators succeed in implementing OSH measures in actual construction projects. The lack of measurable outcomes for their efforts are problematic to both OSH coordinators and other OSH professionals. It is a problem because OSH professionals often perform multiple roles in their organizations, while lacking organizational understanding and recognition of the value of their OSH responsibilities [17]. OSH coordinators may face this same problem, when they experience a lack of compliance from employers and workers, as Rubio et al. describe [13].

However, organizational compliance to safety rules and directions may not even be the most central issue to promote OSH. As Swuste et al. [3] describe, focus on compliance may actually be misguided. The focus on compliance, education, information, and PPEs may actually just be the most readily applicable areas to address rather than targeting more complicated, and potentially expensive, OSH measures. It has been argued that OSH professionals have become too concerned with legitimizing and socializing practices, which serve to show both internally and externally that organizations care about OSH with negligible impact on physical, material, or structural risk factors [18]. Hence, there is a need to study and make visible the impact of OSH professionals on OSH. As OSH coordinators have a prominent position in the efforts to improve OSH in construction work across the European Union, this special group of OSH professionals are suitable for this endeavor.

Organizational OSH and interventions have been investigated both qualitatively and quantitatively, for instance, through safety culture [19,20,21] and safety climate [22,23,24]. However, the impact of OSH professionals has continuously eluded conceptualization. Several studies suggest that this is why OSH professionals struggle in organizations and why their profession lacks professional consensus on methods, practice, ethics, and directions [10,11,25].

In order to provide a conceptualization of measurable impacts of OSH coordinators’, vis-à-vis OSH professionals’, work, this study engages with a large observational material based on a field study with OSH coordinators to suggest and test a methodology to approximate the impact of OSH coordinators’ work in a real-world setting. The study analyzes the field notes from 107 days of observing 12 highly regarded OSH coordinators. During the fieldwork, we found that evaluating each attempt by the OSH coordinators to implement OSH measures and placing it on the *hierarchy of controls* (HOC) for safety prevention showed promise as a way of visualizing the impact of their work. The methodology also quantifies the level of success by identifying whether OSH coordinators manage to either (1) implement OSH measures, or (2) are denied from implementing each measure.

While the study primarily focuses on OSH coordinators and shows some of the reasons why OSH in the construction industry has not been measurably improved despite continuous efforts, its main contribution is to provide methods that enable practitioners, decision-makers, and researchers to better understand the impact of OSH professionals more widely. The study also contributes to insights for OSH coordinators and OSH professionals seeking to improve the legitimacy of their work, and to organizations wanting to ensure the effectiveness of their organizational OSH practices. We hope that this study will set a trend of further research on the impact of OSH professionals’ work, which could lead to more research-based practices, education, and professionalization of OSH professionals.

## 2. Materials and Methods

### 2.1. The Hierarchy of Controls

The HOC has seen increasing attention in recent years and has been recommended by national institutes for occupational safety and health in Australia, Canada, Denmark, and the US [26,27,28,29]. While not being the topic of much scientific interest, the HOC has its roots in what Manuele terms the safety decision hierarchy [30], which was later termed as the hierarchy of controls [31]. The HOC is depicted in Figure 1 and is explained as a five-level hierarchy of effectiveness of safety measures.

### 2.2. Elimination

The elimination of safety risks by designing tasks or processes out of the work process is considered the most effective form of safety work. If a risk-filled process disappears, its associated risk disappears too. The intention in this form of prevention is to minimize human interaction with equipment, materials, and processes containing risk. Thereby, the risk for human mistakes is removed from the work process.

### 2.3. Substitution

Substitution means to reduce risk by using less-hazardous methods, materials, or processes than otherwise planned. This can concern implementing automatic handling of materials rather than manual, substituting dangerous chemicals for less-dangerous chemicals, or replacing machinery with newer and safer alternatives. By making these substitutions, reliance on human actions is reduced, however not as effectively as with elimination.

### 2.4. Engineering Controls

The implementation of safety devices and warning systems built into machinery and equipment is termed engineering controls. These controls are used to separate the worker from hazards and reduce the probability for error. Such controls include guards, automatic locking systems, or startup alarms on equipment, detection systems, safety nets, or fall safety systems.

### 2.5. Administrative Controls

Administrative controls are warning systems, signals, signs, labels, instructions, training, education, or procedures to affect the behavior, reactions, or practice of people. These controls include developing work methods and procedures, selection of personnel, training, guidance, supervision, instruction, planning, motivation, job rotation, change management, behavioral/cultural/practice changes, or inspection. Reaching high levels on all these initiatives is rare.

### 2.6. Personal Protective Equipment (PPE)

Some tasks require appropriate protective equipment. Providing this requires identification of needs, of fitting, training, inspection, and maintenance of this equipment. Examples of providing this include safety shoes, visibility vests, glasses, helmets, and so on. Even though equipment is important, these are the least effective solutions, as they seek to mitigate risks that may be present rather than removing the risks altogether.

In construction OSH, the fundamental logic of the HOC has been shown to align well with the Szymberski curve [32] as eliminations and substitutions may be easier to achieve in early phases of the construction project [33]. Further, both approaches argue that implementing measures of the higher levels of the HOC, i.e., elimination and substitution, more effectively improve OSH than lower level measures such as administrative controls or PPE. This is also the theoretical assumption of this article, that measures implemented at the higher levels of the HOC are more effective and therefore desirable to implement.

In the following analysis, the HOC is used as framework to evaluate the empirical data. In the analysis, we initially identify all situations in which OSH coordinators successfully implemented an OSH measure as well as situations where these attempts to implement an OSH measure was denied. These situations and their associated OSH measure were assessed in relation to the HOC as a way of evaluating the level of safety effectiveness attributable to each measure. This approach has been employed by Lingard et al. [33] to identify the effectiveness of early implementation of safety measures in construction projects.

## 3. Methods

### 3.1. Participants

The analysis is based on data consisting of field notes made by observing 12 experienced OSH coordinators across multiple Danish construction projects. The OSH coordinators were selected using a Delphi-inspired survey [34]. Seventy-nine OSH experts from the Danish construction industry were invited to the survey. The experts were employed in unions, employers’ associations, and professional interest associations, as well as OSH and engineering consultancy companies. They were asked to identify OSH coordinators who were proficient at their jobs. Ninety-five different coordinators were identified. From this list of 95 names, we invited the coordinators who received the most nominations from the top of the list (11 nominations) until we had 12 participants for the study. The 12 participating coordinators all received recommendations from at least two members of the expert panel. From the list of nominated OHS coordinators, 12 other OHS coordinators declined to participate for the following reasons: (1) not currently performing coordinator tasks (*n* = 8), (2) working at confidential construction sites (*n* = 2), (3) retirement (*n* = 1), and lack of time (*n* = 1). All of the invited OHS coordinators expressed great interest in the study.

The 12 participating OSH coordinators were all experienced at OSH coordination and other OSH advisory work in the construction industry. They were between 40 and 70 years of age and had between 14 and 50 years of experience. Eight were male and four were female.

This approach to the selection of “successful coordinators” means that the measure of successfulness is based on recognition among peers. As more technical or objective measures of success for OSH coordinators or OSH professionals in general do not yet exist, this approach was also chosen from the critical perspective that social recognition and positioning in the professional field may not correspond to the actual ability to improve OSH. However, a *critical case* validity still applies in the sense that it can be expected that if these “known to be successful” coordinators experience particular challenges, then other OSH coordinators are likely to experience similar or bigger challenges. This enables generalizability in the sense that if a given factor is a challenge to these successful coordinators, then, most likely, it will also be to others [35].

### 3.2. Observational and Analytical Approach

The 12 OSH coordinators who were nominated and selected to participate in the study were followed for 7–10 workdays. This meant that the researcher followed the OSH coordinator during office work, meetings both on and off the construction site, safety walks and safety meetings, and during every other activity in their general practices as OSH coordinators. In total, the researchers observed 107 workdays which were recorded in extensive field notes. The field notes were further elaborated during breaks under the fieldwork, and on subsequent workdays after the days of fieldwork. The focus for the observations was to investigate how the selected coordinators engage in relations with other actors in their places of work, what practices they conduct to improve or maintain OSH at the assigned construction projects, and what measures they tried to implement. It was our ambition to cover OSH coordination in both design and execution phases of the projects. However, the coordinators’ tasks varied during our observations, and we managed to observe approximately 42 (39.3%) days in the design phase and 65 (60.7%) days in the execution phase. It is difficult to precisely distinguish between the two, as the tasks of the coordinators would often vary during the day.

For this study, all field notes were imported to NVivo 11 [36], and were initially coded into two categories: (1) situations in which the OSH coordinator successfully carries through or implements an OSH measure according to any of the HOC categories, and (2) situations in which the OSH coordinator is denied in carrying through or implementing an OSH measure according to any of the HOC categories. Following this initial procedure, each of the initially coded situations were evaluated by two independent researchers in relation to the five HOC categories, (1) elimination, (2) substitution, (3) engineering controls, (4) administrative controls, and 5) personal protective equipment, in order to make an assessment of the expected effectiveness based on the OSH measure. In situations of disagreement, the researchers discussed evaluations based on the HOC and agreed upon the most suitable categorization. As is apparent from this description, the HOC-lens was not applied to the field work until the following analysis. This means that the field work was not entirely structured to fit with the HOC analysis. The consequences and limitations connected with this approach are addressed in the discussion.

Being based on a field work study, one might expect more lengthy qualitative analyses. These, however, are reported in other work concerning the practices [37] and the professional identities [38] of OSH coordinators. This study is particularly concerned with conceptualizing the HOC as a means for mapping the measures implemented by the OSH coordinators and, as such, are reported in a short and rather quantitative form.

## 4. Results

The analysis revealed that during the 107 observation days, the 12 coordinators implemented 280 OSH measures; the average for all coordinators was 23.3, the median was 25.5, and the top and bottom outliers were 38 and 8. In total, the 12 coordinators were denied from implementing measures that they sought to initiate 71 times; the coordinators were denied 5.9 times on average, the median was 6, and the top and bottom outliers were 20 and 2.

### 4.1. Implemented Measures

The 280 successfully implemented measures were characterized according to the HOC model, and the results are shown in Figure 2.

As Figure 2 shows, the main bulk of the implemented measures belongs to the categories *administrative controls* (53.9%) and *engineering controls* (35%). In the following, we have broken down the implemented measures into more detailed subcategories under each of the HOC levels—where these were applicable. In the tables below, the implemented measures are visualized. When we categorize a measure as implemented by the OSH coordinator, this does not mean that the coordinator managed or carried out the implementation on their own. Rather, it means that the OSH coordinator, during our observations, verbally or in writing, agitated for the implementation of the particular measure, and that it was subsequently carried out or put into action. Usually, this agitation would take place in the sense that the OSH coordinator performed a safety walk, held a safety meeting, or examined the design material. Based on this, the coordinators would contact relevant parties (e.g., engineer, supervisor, site manager, project manager, worker, etc.) and discuss the particular measure, which they found reasonable to address.

### 4.2. Eliminations

As shown in Figure 2, the coordinators implemented two eliminations. The two *eliminations* (0.7%) both concerned instances where a coordinator, during the planning phase of a project, managed to negotiate for a sanitation of lead and polychlorinated biphenyls (PBC) before allowing workers into the construction area. The research group considered these to be eliminations, because the workers, who are not equipped with the tools, protection, or education to handle these substances, were not exposed to them. Instead, professional sanitation companies took care of this.

### 4.3. Substitutions

*Substitution* accounted for 2.9% of the implemented measures; here, five out of eight measures concerned switching from manual transport or lifting to some form of machine-assisted transport and procedure. In Table 1, the types of measures that the coordinator in some way contributed to implementing are categorized:

### 4.4. Engineering Controls

Within engineering controls (35%), changes to the walking paths, measures separating driving and walking traffic, and measures improving safety railing were prominent. Asking supervisors, contractors, and workers to move their materials from access paths was another relatively frequent occurrence. In addition, participating in logistic planning in both the design phase and the execution phase was something that coordinators relatively often took part in. Writing specific procedures concerning crane work, ways to handle contaminated dirt and separation between workers on lower floors and montage work on higher floors were design-phase-oriented activities in this category. Table 2 specifies the measures implemented by the observed OSH coordinators in the engineering controls category:

### 4.5. Administrative Controls

Within administrative controls (53.6%), most of the measures included safety meetings, introductions, and workshop activities where coordinators discussed the production with supervisors, managers, and workers. In addition, safety walks on the construction site were a prominent activity. These measures, which were primarily concerned with talking about safety, were scored as an independent activity, so in cases where they led to other measures, that was scored separately. Table 3 shows the administrative controls implemented during our observations.

### 4.6. Personal Protective Equipment

*PPE* accounted for 7.5% of the implemented measures. Here, coordinators getting workers to put on helmets were the most occurring activity. Only on rather rare occasions did the coordinators address or implement other types of PPE. The implemented measures in the PPE category are displayed in Table 4 below.

## 5. Denied Measures

The distribution of the 71 denied initiatives on the HOC are shown in Figure 3. Similar to the implemented measures, the denied measures are often based on situations where the OSH coordinators had performed a safety walk, meeting, or examination of the design material. In most of these cases, the coordinator would address the OSH issue, and in some form sustain a verbal rejection of their suggestions. In other cases, rejections would come as, for instance, a shrug or a person just saying “yes” and not doing anything about the issue.

Of the measures that coordinators sought to implement but were denied, *administrative* controls (50.7%) and *engineering controls* (36.6%) again measured the highest number of instances. In the following, the denied initiatives are broken down into subcategories.

### 5.1. Eliminations

The three *eliminations* (4.2%) concerned a coordinator denied from implementing a sanitation of lead on the construction site, a coordinator denied from specifying general demands of health and safety in the projecting materials, and a coordinator denied from requesting a silent cooling system instead of a noisy one, requiring noise protection to be nearby.

### 5.2. Substitutions

The two *substitutions* (2.8%) concerned a coordinator being denied a noise-dampening shed for cutting rebar and other forms of iron, and a coordinator being denied changing the working procedure for mounting concrete floors from being based on drilling to using prefabricated inserts.

### 5.3. Engineering Controls

For engineering controls (36.7%), people refusing to clear up trash or materials in access paths and people refusing to build railings were the most frequent. The 26 denied engineering control initiatives are summed up in Table 5.

### 5.4. Administrative Controls

Among the denied administrative controls (50.7%), being denied from specifying OSH management procedures in the design material were the most prominent. People refusing to participate in safety meetings, not showing up as agreed, and not allowing their employees to participate were also prominent measures. The administrative control initiatives that were denied are displayed in Table 6:

### 5.5. Personal Protective Equipment

For *PPE* (5.6%), two instances regarded workers refusing to use safety lines when working close to edges, one worker refused to wear a helmet, and one refused to wear glasses.

In the following discussion, the meaning and implications of the analysis are discussed and interpreted in the light of existing research.

## 6. Discussion

This study is the first to engage with the impact of OSH coordination from an empirical perspective. In this regard, we find it thought-provoking that the overwhelming bulk of implemented measures are categorized within the lower levels of effectiveness in the HOC. On one hand, this indicates that successful coordinators pay great attention to communicational practices that may be predecessors for the implementation of other measures, as has been pointed out as important from numerous studies [11,14,39]. However, while safety meetings, walks, and separate traffic paths are important, this may also mean that the implemented measures are not sufficient to improve OSH in general. As initially discussed, designing for elimination or substitution of unsafe working methods is generally considered most effective [3,33,40]. In addition, it has actually recently been shown that communicational practices and strategies may even have adverse effects on safety [41].

One reason why implemented measures may be mainly reserved for administrative or engineering controls could be understood through the suggestion made by both Rubio et al. [13] and Swuste et al. [3], that compliance is harder to obtain than engagement, and may actually divert attention from harder and more complex preventive measures. This may also partially explain why the coordinators generally seem to mainly focus on these activities, which could be interpreted from the fact that they are not very often denied from higher level initiatives either. It is likely that these successful and experienced coordinators have a reasonably well-developed sense of the room for agency available to them, and therefore do not suggest many initiatives outside usual practice. In this regard, it is interesting that OSH coordinators were not observed to be denied regarding technical assistive devices even one single time, which is actually the type of substitution they most often manage to implement. This means that perhaps implementing more technical assistive devices may be an area of future concern for OSH coordinators.

Another particularly interesting aspect of the study is the point that, within the denied administrative controls, suggesting specific management procedures for OSH in the design material is actually the most frequently occurring observation. This category concerned situations where the coordinator suggested plans, procedures, or software platforms for managing OSH, which may professionalize and standardize ways of engaging with OSH in the construction organization. This is also the only area in which coordinators manage to implement fewer initiatives than they are denied. As such, it may be that coordinators view this as a highly important area to gain traction within the organization.

In any case, OSH coordinators do not manage to implement a very high proportion of initiatives in the higher levels of the HOC, which means that their work may not be as efficient in improving OSH as it could be. In this sense, this study provides an empirical exemplification of the concerns raised by Rae and Provan [18] that OSH professionals are too concerned with legitimization and socializing practices, and not enough with material or structural prevention of risks. This is an important contribution to understanding why OSH in construction has not been measurably improved despite continuous efforts. It is our suggestion that OSH coordinators and professionals may employ this methodology as a way of analyzing their work to reflect on their own practices and investigate the need to improve these, i.e., improve the effectiveness of OSH improvement interventions in organizations which ultimately will benefit the organizations as well. In addition, this may be important to engage organizations in practices that address the lower levels of the HOC, because creating a visibility of this issue may provide a tool for the OSH coordinator and thereby improve their bargaining power. On the other hand, organizations that are interested in improving OSH may also concern themselves with what initiatives on the HOC are implemented in their construction projects.

Improving OSH on construction sites by enabling OSH coordinators to successfully negotiate for eliminations and substitutions may be a fruitful direction to pursue for both researchers and for the OSH in the construction industry. This may be highly relevant for both construction owners who wish to ensure OSH on their projects, for politicians wishing to evaluate the effect of OSH coordination as a political initiative, and for OSH coordinators and advisors wishing to improve their practice and effectiveness.

On a broader notion, we do suggest that applying the HOC to the analysis of other fields of OSH rather than to accident prevention specifically, as has been customary [30,31,42], could be fruitful. As Manuele [30] notes, “a good number of safety professionals do not have in place, systems to determine whether the actions taken in accord with their recommendations achieve the risk reduction intended (p. 186)”. Most often, research and initiatives within the psychosocial working environment are focused on implementing measures that fit within the category of administrative controls (e.g., communication plans, core values, social recognition, etc.). Applying the HOC to this field may reveal that many measures do not address the underlying risk factors that may, for instance, be related directly to work load, time pressure, social exposures, or other risk factors. In addition, this could prove a step in the direction of increasing transparency and impact assessment of safety professionals’ work, which has been identified as a problematic issue in recent research [11]. As such, the HOC may be beneficial to improving OSH more broadly by moving focus from topics and measures that are convenient and easy to talk about to topics or measures that may actually be effective. We would not suggest to remove or lessen focus on the administrative controls and communication regarding OSH, as this could divert attention away from OSH altogether. However, we would suggest that directing a focus on the effect of conversation and administrative controls on the implementation of measures on the other levels would be reasonable to aim for.

A few additional methodological reflections seem appropriate. The methodology employed for the selection of OSH coordinators in the present study confers a particular focus on coordinators who have been successful in making themselves known or positively recognized among colleagues in expert positions. This does not necessarily mean that these are actually more prolific at achieving the implementation of OSH measures. However, the knowledge produced by this perspective reveals the level of implemented measures among generally well-respected OSH coordinators within the Danish field of construction work. As such, even though there may be many other competent OSH coordinators out there, we would not expect to achieve higher success rates by choosing a different sample. This is also emphasized by the age and years of experience obtained by these studied coordinators. On the contrary, we may indeed expect that younger and more inexperienced coordinators would achieve fewer implemented measures. Because the larger part of coordinators in the study were male, we did make an estimation of whether there were differences in the number of implemented measures based on the gender of the coordinators. This did not seem to be the case.

The conceptualizing character of this work is another important factor to be accounted for. While the researchers conducting the fieldwork were focused on OSH measures implemented by the coordinators, they did not perform the fieldwork from the perspective of the HOC. This means that the sub-themes to the HOC are not particularly standardized. On the positive side, this means that measures that might not have fit within predefined categories could still be evaluated. On the negative side, a standardized observation guide might have shown more measures, which could, in this study, have evaded the researchers as being “trivial” in comparison to other relational or material practices taking place simultaneously. We do, however, believe that this was minimal. As there are no previous studies on this topic to compare to, it is hard to determine whether the coordinators under investigation manage to implement a high number of OSH measures or not. We suggest that future research may take this up to offer more insight in this direction. In addition, studies engaging more qualitatively with the situations leading to the implementation of measures or studies investigating and benchmarking the measures associated with different coordinator practices or organizational approaches may be highly interesting to pursue in the future.

The HOC offers a somewhat broad categorization of measures, which is ideal for gaining an abstract overview of what type of measures are both implemented and denied. The strength of this lies in the offer of systematizing knowledge from observations; however, if one is interested in delving further into microsociological or contextual aspects of OSH coordinators’ work, other methods must be consulted and employed. The HOC is also limited in the sense that it lacks a conceptualization for certain kinds of measures. Standardized guides for this may be developed; however, we suggest maintaining a focus on general principles, since this allows for reframing and reflecting on the measures and subcategories fitting within each level on a contextual basis. This may be a better approach than defining narrow subcategories that may be increasingly subject to gaming and new legitimization practices.

## Figures and Tables

**Figure 1 ijerph-19-02731-f001:**
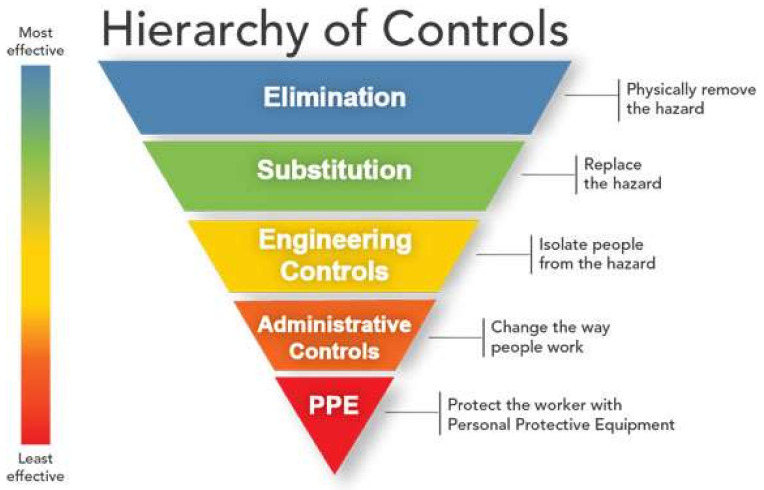
NIOSH model of HOC [28].

**Figure 2 ijerph-19-02731-f002:**
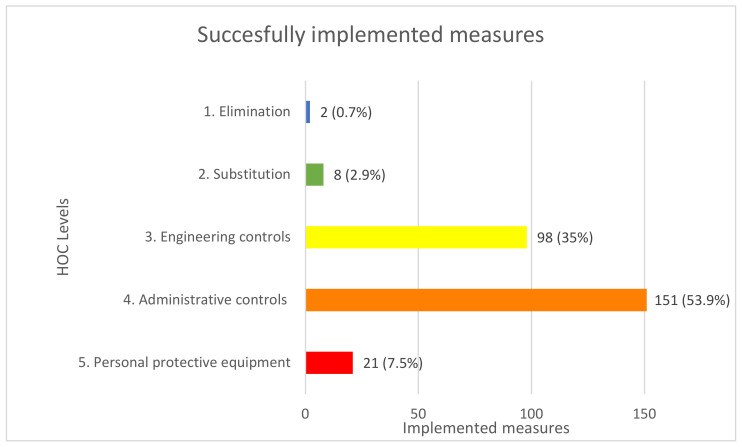
Number of implemented measures and percentage of HOC measures.

**Figure 3 ijerph-19-02731-f003:**
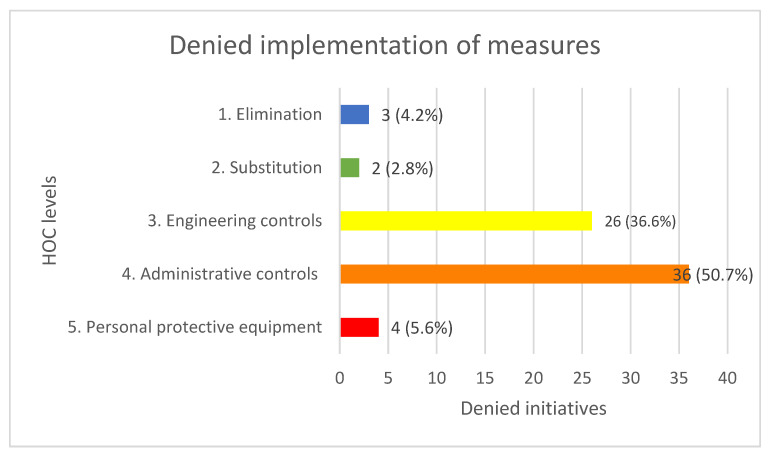
Number of denied initiatives and percentage of denied HOC measures.

**Table 1 ijerph-19-02731-t001:** Subcategories of successfully implemented *substitution* measures.

Type of Implemented *Substitutions*	No. of Implemented Measures
Use of technical assistive devices	5
Alternative chemical product	1
Ergonomic space for tying rebar on site	1
Platform for materials instead of in-building transport	1

**Table 2 ijerph-19-02731-t002:** Subcategories of successfully implemented *engineering controls*.

Type of Implemented *Engineering Controls*	No. of Implemented Measures
Improved access paths	23
Safety railing	21
Logistic plan	12
Move materials from access paths	11
Procedures for crane work, handling poisoned dirt or concrete montage in design material	9
Protective cape on rebar, tying fire extinguishers, replacement of worn electrical wires	7
Proper excavation degrees or digging boxes	5
Safety gear or dust suction on machines	4
Lighting on paths	3
Specific scaffolding for tasks	2
Improve column safety	1

**Table 3 ijerph-19-02731-t003:** Subcategories of successfully implemented *administrative controls*.

Type of Implemented *Administrative Controls*	No. of Implemented Measures
Safety meeting	44
Perform safety walk	20
Revise PSS or risk log	13
Facilitate safety induction	11
Facilitate startup or process workshop	8
Demand risk assessment from entrepreneur	7
Discuss safety at other meeting	7
Specify OSH management procedure in design material	6
Specify communication plan in design material	6
Tutor a colleague in risk assessment or other OSH	5
Updated visual construction site plan on site	5
Put up safety signs	5
Plan social events and external communication (e.g., with municipalities or citizens)	4
Plan helmet/noise protection campaign	4
Conduct design phase to execution phase transition meeting	4
Near-miss accident evaluation meeting	2

**Table 4 ijerph-19-02731-t004:** Subcategories of successfully implemented *personal protective equipment*.

Type of Implemented *Personal Protective Equipment*	No. of Implemented Measures
Use of helmets	10
Use of fall equipment	3
Use of noise protection	3
Use of sunscreen	1
Use of safety glasses	1
Use of safety shoes	1
Use of safety masks	1
Use of fire extinguisher nearby	1

**Table 5 ijerph-19-02731-t005:** Subcategories of denied *engineering controls*.

Type of Denied *Engineering Controls*	No. of Denied Initiatives
Removing materials from access paths	5
Safety railing	5
Improving access paths	4
Logistic plan	3
Safety gear or dust suction on machines	2
Protective cape on rebar, tying fire extinguishers, replacement of worn electrical wires	2
Proper excavation degrees or digging boxes	2
Specific scaffolding for tasks	2
Lighting on paths	1

**Table 6 ijerph-19-02731-t006:** Subcategories of denied *administrative controls*.

Type of Denied *Administrative Controls*	No. of Denied Initiatives
Specify OSH management procedure in design material	11
Safety meeting	8
Facilitate safety introduction	4
OSH engagement initiative	3
Specify communication plan in design material	3
Facilitate startup or process workshop	1
Near-miss accident analysis	1
Helmet campaign	1
Demand risk assessment	1
Truck certificate	1
Alcohol policy	1
Conduct design phase to execution phase transition meeting	1

## Data Availability

Data are held at The National Research Centre for the Working Environment in Denmark. Data may be obtained by contacting the main author jza@nfa.dk and taking applicable GDPR laws into account.

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
