# Peer review of "The Hierarchy of Controls as an Approach to Visualize the Impact of Occupational Safety and Health Coordination"

_ijerph, 2022, doi:10.3390/ijerph19052731_

Round 1
Reviewer 1 Report
This paper conceptualizes the Hierarchy of controls (HOC) as a means for visualizing and evaluating the impact of OSH coordinators’ work. The results are interesting and corroborative. Overall, this work is suitable for publication on the current journal. However, minor revisions are required before publications.
- The authors should highlight their main contributions in the introduction part.
- The authors should present a discussion about their further study.
Author Response
Comments and Suggestions for Authors
This paper conceptualizes the Hierarchy of controls (HOC) as a means for visualizing and evaluating the impact of OSH coordinators’ work. The results are interesting and corroborative. Overall, this work is suitable for publication on the current journal. However, minor revisions are required before publications.
We sincerely thank the reviewer for their time in reading the manuscript and for their positive assessment!
The authors should highlight their main contributions in the introduction part.
We sincerely thank the reviwer for pointing this out. There was definitely room for improving the specification of the study’s contributions. Therefore we have rephrased the contribution part of the introduction to the following:
While the study primarily focuses on OSH coordinators and shows some of the reasons why OSH in the construction industry has not been measurably improved despite continuous efforts, its main contribution is to provide methods that enables practitioners, decision makers and researchers to better understand the impact of OSH professionals more widely. The study also contribute with insights for OSH coordinators and OSH professionals seeking to improve the legitimacy of their work, and to organizations wanting to ensure the effectiveness of their organizational OSH practices. We hope that, this study will set a trend of further research on the impact of OSH professionals’ work, which could lead to more research-based practices, education and professionalization of OSH professionals.
The authors should present a discussion about their further study.
We agree that further suggestions for future studies could be added and have made the following reflections in the discussion section.
Also, studies engaging more qualitatively with the situations leading to the implementation of measures or studies investigating and benchmarking the measures associated with different coordinator practices or organizational approaches may be highly interesting to pursue in the future.
Reviewer 2 Report
Recently, occupational safety and health check have been very important matter. This paper mentions about the problems by the point of workplace hierarchy system and the contents are very interesting, too!
However, if you could, please add below points.
- Please add the special features of the 12 successful OSH coordinators in detail. What were the special points to become a successful OSH coordinators?
- Good and efficient work place does not need administrative controls so much. How do you plant to reduce the administrative controls? If you could please add communication dimension.
Author Response
Comments and Suggestions for Authors
Recently, occupational safety and health check have been very important matter. This paper mentions about the problems by the point of workplace hierarchy system and the contents are very interesting, too!
We thank the reviewer for their time and appreciate these constructive and positive comments.
However, if you could, please add below points.
Please add the special features of the 12 successful OSH coordinators in detail. What were the special points to become a successful OSH coordinators?
We have sought to clarify this in the methods section. In order to qualify as a successful coordinator, one had to be selected by peers. So the means of being successful in this study is to have positioned oneself in a way which made their peers recommend them as good at their jobs. We specified the selection of these in the methods section:
OHS coordinators were selected using a Delphi-inspired survey 31 among 79 OHS experts in the industry. This expert panel consisted of people employed in unions, employers’ associations as well as OHS advisory companies. They were asked to identify up to six OHS coordinators who were good at their jobs. The experts nominated 95 different coordinators. From this list of 95 names, we invited the coordinators who received nominations from most experts (11 nominations) until we had 12 participants for the study. The 12 participating coordinators all received at least three nominations for being good at their job from the expert panel.
Good and efficient work place does not need administrative controls so much. How do you plant to reduce the administrative controls? If you could please add communication dimension.
This is a good point of discussion. While we would not suggest to remove or lessen communication, we would suggest that directing a focus on the effect of conversation and administrative controls on the implementation of measures on the other levels would be reasonable to aim for. Hence we added the following to the discussion:
We would not suggest to remove or lessen focus on the administrative controls and communication about OSH, as this could divert attention away from OSH altogether. However, we would suggest that directing a focus on the effect of conversation and administrative controls on the implementation of measures on the other levels would be reasonable to aim for.
Reviewer 3 Report
Comments for Authors
Journal: IJERPH (ISSN 1660-4601)
Manuscript Number: ijerph-1575393
The Hierarchy of Controls as an approach to visualize the impact of occupational safety and health coordinators work?
Overall this is a well-organized and interesting work addressing a vital problem in the construction industry. The article makes a valuable contribution to the state-of-the-art knowledge on improving Occupational safety and health in construction. However, certain points of concern need to be addressed before this paper is ready for publication.
- In the introduction, the section should provide statistics on various challenges faced by the construction industry due to poor OSH. Particularly give an amount of expenditure associated with covering OSH incidents.
- In addition, in introduction there is need to distinguish between OSH coordinators and OSH professionals. Sometimes these terms are used interchangeably, while at other locations, they are defined in two separate positions.
- The methodology part relating to coordinators' selection is a bit confusing. Did you initially shortlist 24, and then 12 rejected the survey? Please clarify in the manuscript.
- The results and discussion sections are well written, but the impacts of coordinators' gender diversity and years of experience are missing. There is vital need relate to the measures to genders and years of experience so resources are effectively directed.
- The paper is well written but there are several language errors. I would recommend a good editorial review before submission.
- Quality of all figures needs to be improved.
- Figure 2 and 3 are missing caption titles. Axis titles need to be added. In addition to the amount write percentage of HOC measure. The colors in Figure 1 can be matched with associated bars in Figure 2 and 3. Preferably add patterns and legend.
- Text in tables should be left aligned to improve readability. Please provide proper caption headings to tables along with related column headings. Some of the measures look like column headings and are in bolt.
- Follow citation style of the International Journal of Environmental Research and Public. You seem to have used two different and incorrect styles (e.g. Page 1, Line 32 and Page 1, Line 38)
- Some minor comments are
- Page 2, line 46 is it “cooperation between employers” or “cooperation between employers and employees”?
- Page 5, line 221 and 222 (marked as “XX”) are missing references?
- Terms with similar words e.g. Page 5, line 219, “are discussed in the discussion,” should be avoided.
- Page 13, line 470. The year of Møllers’ work (The competences of successful safety and health coordinators in construction projects) is 2021, not 2020. Please verify remaining references are correct.
- Make sure all captions are properly titled and referred in text.
Author Response
The Hierarchy of Controls as an approach to visualize the impact of occupational safety and health coordinators work?
Overall this is a well-organized and interesting work addressing a vital problem in the construction industry. The article makes a valuable contribution to the state-of-the-art knowledge on improving Occupational safety and health in construction. However, certain points of concern need to be addressed before this paper is ready for publication.
Initially, we would like to thank the reviewer for their time and effort in providing insightful and constructive comments to our work! We have worked intensely with revising the work.
In the introduction, the section should provide statistics on various challenges faced by the construction industry due to poor OSH. Particularly give an amount of expenditure associated with covering OSH incidents.
We appreciate, that specifying the costs and particular risk associated with construction work contributes to establishing the burning platform of our study. Therefore we changed the introductory section to this:
Occupational illness and accidents in the European Union costs more than 475 billion Euros each year1. Since the 19th century2, the construction industry has been, and continues to be, one of the most hazardous industries to work in. Occupational safety and health (OSH) threats, such as accident and physically exerting work, still challenge management and OSH initiatives today3–5. These observations are emphasized by Eurostat showing that 716 fatalities and 371,732 non-fatal accidents were registered in construction in 2016, and that the fatality rate in construction was 3.4 times higher than the EU average for 2016, and 1.9 times higher for nonfatal accidents6.
In addition, in introduction there is need to distinguish between OSH coordinators and OSH professionals. Sometimes these terms are used interchangeably, while at other locations, they are defined in two separate positions.
We agree, that there is a need to specify, that OSH coordinators are just one type of OSH professional among many. In order to accommodate this suggestion we added the following to the introduction:
One particularly important political initiative in seeking to improve OSH in the construction industry is health and safety coordination (OSH coordination). OSH coordination is undertaken by OSH coordinators - a particular type of OSH professional9,10 responsible representing the construction client in matters of OSH during the project. OSH coordination has since its introduction in EU-legislation through directive 92/57/EEC in 1992, become a prime tool in political initiatives seeking to improve health and safety among construction workers. In brief, the OSH coordinator is appointed to coordinate health and safety at sites with more than one employer present11. The OSH coordinator is responsible for ensuring that employers apply the general prevention principles and for ensuring the cooperation between employers in matters of OSH.
The methodology part relating to coordinators' selection is a bit confusing. Did you initially shortlist 24, and then 12 rejected the survey? Please clarify in the manuscript.
Agree that this could be clearer. We have sought to clarify the recruiting procedure in the study by specifying the following.
The OSH coordinators were selected using a Delphi-inspired survey34. Seventy-nine OSH experts from the the Danish construction industry were invited to the survey. The experts were employed in unions, employers’ associations, professional interest associations, as well as OSH and engineering consultancy companies. They were asked to identify OSH coordinators who were proficient at their jobs. Ninety-five different coordinators were identified. From this list of 95 names, we invited the coordinators who received most nominations from the top of the list (11 nominations) until we had 12 participants for the study. The 12 participating coordinators all received recommendations from at least two members of the expert panel. From the list of nominated OHS coordinators, 12 other OHS coordinators declined to participate for the following reasons; 1) not currently performing coordinator tasks (n=8), 2) working at confidential construction sites (n=2), 3) retirement (n=1), and lack of time (n=1). All of the invited OHS coordinators expressed great interest in the study.
The results and discussion sections are well written, but the impacts of coordinators' gender diversity and years of experience are missing. There is vital need relate to the measures to genders and years of experience so resources are effectively directed.
We agree that these are valid and important topics to touch upon and have added the following to the discussion:
As such even though there may be many other competent OSH coordinators out there, we would not expect to achieve higher success rates by choosing a different sample. This is also emphasized by the age and years of experience obtained by these studied coordinators. On the contrary, we may indeed expect that younger and more inexperienced coordinators would achieve fewer implemented measures. Because the main part of coordinators in the study were male, we did make an estimation of whether there were differences in the number of implemented measures based on the gender of the coordinators. This did not seem to be the case.
The paper is well written but there are several language errors. I would recommend a good editorial review before submission.
In our revision of the work, we have performed an extra careful proof-reading of the manuscript.
Quality of all figures needs to be improved.
Figure 2 and 3 are missing caption titles. Axis titles need to be added. In addition to the amount write percentage of HOC measure. The colors in Figure 1 can be matched with associated bars in Figure 2 and 3. Preferably add patterns and legend.
Text in tables should be left aligned to improve readability. Please provide proper caption headings to tables along with related column headings. Some of the measures look like column headings and are in bolt.
We thank the reviewer for pointing these issues with figures out. Therefore, we also added titles underneath each figure. We added axis titles and percentages. Indeed we also took to heart the idea of matching colors, which makes for a very nice and vibrant visual effect of the paper. Additionally, we revised the tables to accommodate the coloration scheme as well, and adjusted tables in accordance to these suggestions.
Follow citation style of the International Journal of Environmental Research and Public. You seem to have used two different and incorrect styles (e.g. Page 1, Line 32 and Page 1, Line 38)
We realize there were a number of similar mistakes, which we have corrected. This was also done as part of the extra thorough proof-reading which we have performed.
Some minor comments are
Page 2, line 46 is it “cooperation between employers” or “cooperation between employers and employees”?
We thank the reviewer for their thoroughness in addressing this. It is ‘cooperation between employers’ because the coordinator must ensure cooperation on construction sites where several employers are often present.
Page 5, line 221 and 222 (marked as “XX”) are missing references?
These are articles currently under review. We have added these references.
Terms with similar words e.g. Page 5, line 219, “are discussed in the discussion,” should be avoided.
We realize this narrative blunder. We have changed to ‘are addressed in the discussion’.
Page 13, line 470. The year of Møllers’ work (The competences of successful safety and health coordinators in construction projects) is 2021, not 2020. Please verify remaining references are correct.
Make sure all captions are properly titled and referred in text.
We thank you for correcting us in this mistake. We have corrected the publication year of this publication. Also we did, as part of our general proofreading go through all the references to adjust any mistakes.
Round 2
Reviewer 3 Report
All my comments have been addressed in a proper way. From my side, the paper is now ready for publication. Thanks for appreciating our input, this gives new motivation for reviewing!